# Dynamic Evaluation of Ecological Service Function Value of Qilihai Wetland in Tianjin

**DOI:** 10.3390/ijerph17197108

**Published:** 2020-09-28

**Authors:** Chen Chen, ChaoFeng Shao, YanMin Shi

**Affiliations:** 1National & Local Joint Engineering Research Center on Biomass Resource Utilization, College of Environmental Science and Engineering, Nankai University, Tianjin 300350, China; 2120190566@mail.nankai.edu.cn; 2College of Life Science, Huaibei Normal University, Huaibei 235000, China; ymshi1994@chnu.edu.cn

**Keywords:** land use change, wetland ecosystem, ecosystem services

## Abstract

This study, using the method of economic quantitative analysis, studied the land use changes of Qilihai Wetland from 2008 to 2017, and the effects of these changes on the ES (ecosystem service) values of the wetland. This article benchmarked the 2030 Agenda for Sustainable Development, based on the systematic analysis and analysis of current ecological service function value evaluation methods. The research results show that the total values of the wetland ecosystem services in 2008, 2011, 2014, and 2017 were 317 million yuan, 299 million yuan, 283 million yuan, and 321 million yuan, respectively. In 2008 and 2011, the ES of the Wetland was mainly based on supply and support services, and in 2014 and 2017, it was mainly based on supply and regulation services. Changes in human utilization, natural conditions, and social economy will all lead to changes in the ES value of the whole Wetland. This research can enable decision makers to intuitively understand its ecological changes and plan the use of land and formulate ecological protection measures in a reasonable and effective manner. Finally, the article puts forward relevant suggestions for sustainable development based on the resource and environmental foundation and characteristics of Qilihai Wetland.

## 1. Introduction

Wetland is a transitional zone between terrestrial and aquatic ecosystems, known as the “Kidney of the earth”. The definition of wetland in the Ramsar Convention is considered to be the more authoritative concept of wetland “Wetlands are areas of marsh, fen, peatland or water, whether natural or artificial, permanent or temporary, with water that is static or flowing, fresh, brackish or salt, including areas of marine water the depth of which at low tide does not exceed six metres” [1]. In order to strengthen the management of wetland protection and fulfill the International Wetland Convention, China formulated regulations on Wetland Protection and Management in 2013 and revised them in 2017. The revised term “wetland” refers to “perennial or seasonal water areas, water areas, and sea areas with a water depth of not more than 6 m at low tide, including natural wetlands such as swamp wetlands, lake wetlands, river wetlands, coastal wetlands, and key protected wildlife Artificial wetlands such as habitats or key protected wild plant native sites”. The numerous wild animal and plant resources in the wetland make it one of the most biologically diverse ecological landscapes in nature and the most valuable ecosystem in the world. The value of ecological benefits per unit area is significantly higher than other ecosystems [2]. The second national survey of wetland resources shows that China’s wetland area is 53.62 million hectares, accounting for 5.58% of the country’s total wetland area. China’s wetland area accounts for 4.4% of the global wetland area [3]. Compared with the results of the first survey 10 years ago, the total area of wetlands in China under the same caliber has decreased by 3,396,300 hectares, a decrease rate of 8.82%. Although the protected area of wetlands has been further improved, the threatened pressure on wetlands continues to increase. In addition to natural factors such as climate change, human activities occupying and changing wetland uses are the main reasons for the substantial reduction in wetland area and the increase in threatened pressure [4]. Urbanization and the conversion of natural areas to farmland are one of the most important threats to the value of wetland ecosystems [5]. The severity of these effects depends on many factors, such as the fragility of the ecosystem, the severity and extent of changes, and the sensitivity of plants and animals. Wetlands with a variety of aquatic and plant species are one of the most vulnerable ecosystems, especially for land use changes [6]. The temporal and spatial differences of human-induced land use changes and climate change lead to regional differences in ecosystem services. Science Assessing changes in the value of wetland ecosystem service values and the influencing factors has important theoretical and practical significance for further understanding the evolution of wetland ecosystem service values. Further, it can provide a basis for the implementation of the ecological protection compensation system and the management of wetland parks.

The value evaluation of ecosystem services is a hotspot in the intersection of ecology and geography. Many scholars around the world have explored and researched evaluation methods [7]. In the 1990s, Contanza classified the global ecosystem service functions and classified them into 17 categories [8]. In order to promote the protection and sustainable development of ecosystems, the United Nations has successively issued the Millennium Ecosystem Assessment (MA) Assessment Report and the Global Biodiversity and Ecosystem Services Assessment Report. They emphasize the importance of the benefits of nature to sustainability. However, 60% of the ES involved in the MA are degrading or being used in an unsustainable manner by humans [9]. Many ESs are degraded in order to increase the provision of other services, such as food production, and the costs of degradation are often passed on to another group of people or deferred to future generations in trade-offs between benefits. This degradation of ecosystem services has become a major obstacle to the Millennium Development Goals (MDGs), and as a result, an increasing number of scholars, both domestic and international, are focusing on an integrated, multi-scale assessment of ecosystems around the world, and the importance of ES is increasingly being emphasized. The value of ES reflects not only changes in the physical stock of ecosystems, but also human estimates of the value of the ecosystem sessions consumed in a given socio-economic context, based on specific economic and social development. Based on the existing achievements of MDGs, the United Nations continues to guide the global sustainable development of more comprehensive and specific 2030 Sustainable Development Goals (SDGs) [10]. The Sustainable Development Goals 6 and 14 proposed by the United Nations are related to water, including “protection and restoration of water-related ecosystems.” A huge challenge currently facing the management of water-related ecosystems is that it is related to many human behaviors and environmental factors [11]. Therefore, it is of great significance to study the impact of human behavior on ecosystem changes. If relevant laws and management regulations can be formulated to maintain ecosystem functions or rebuild degraded ecosystems based on local actual conditions, then achieving the relevant sustainable development goals will be much easier.

Nowadays, quantitative assessment of ecosystem service functions using geographic information on the basis of SDGs is also an important way to realize the advancement and implementation of the United Nations 2030 Agenda for Sustainable Development [12]. Research on ecosystem service function assessment in China started late and mainly focused on static and dynamic studies of ecosystems to assess the value of ecosystem service functions [13,14]. The results all show that wetland ecosystems have important functions and great economic value. Among them, dynamic evaluation is mostly studied in combination with land use change [15], which can better understand the reasons for the decline of ES and help to formulate reasonable and effective conservation policies [16]. In recent years, the ecological environment of Chinese wetlands has deteriorated and their functions have declined. For wetland conservation and restoration, China has proposed the establishment of a wetland conservation and restoration system, issued the National Wetland Conservation Thirteenth Five-Year Plan, the Arable Grasslands, Rivers and Lakes Restoration Plan (2016–2030), and a series of programs such as the delineation of wetland ecological red lines and the construction of wetland parks. In addition to their enormous direct economic benefits, wetlands also have obvious ecological and social benefits. While maintaining economic growth, settling this “ecological account” plays an important role in maintaining the environmental support capacity and sustainable development of wetlands. This paper takes Tianjin Qilihai National Wetland Park as an example and uses shadow engineering and alternative cost methods to dynamically assess the ES value of Qilihai wetland in 2008, 2011, 2014, and 2017; quantify the changes in the wetland ecosystem and the dynamic evolution of the service value; and identify and analyze the impact of different policies and conservation actions on the ecosystem.

## 2. Materials and Methods

### 2.1. Study Area

On 27 October 1992, the State Council approved the “Request for Approval of the New National Nature Reserve in 1992” submitted by the Environmental Protection Agency (Guo Han [1992] No. 166), and approved the establishment of Tianjin Ancient Coast and Wetland National Nature Reserve, with a total area of 359.13 km^2^, 270.64 km^2^ of experimental area, 43.34 km^2^ of buffer zone, and 45.15 km^2^ of core area. The reserve consists of shell dike, oyster reef and Qilihai wetland, located in the eastern part of Tianjin region; the geographical coordinates are between 117°26′–117°39′ East longitude and 39°16′–39°20′ North latitude. The protected area is rich in natural resources, with an intact wetland system and high biodiversity. Therefore, the government invested 1 billion yuan in the Qilihai Wetland to build the Qilihai National Wetland Park, a state-level nature conservation area, which officially opened for visitors in 2013. In 2014, the protected area was included in the city’s ecological land protection red line by the Tianjin Municipal Government, which protected all of this rare land [17]. According to the Tianjin Management Regulations for Permanently Protected Ecological Zones, tourism activities are not allowed within the red line of permanently protected ecological zones, so the Qilihai Wetland Park was closed in September 2015 and has been closed ever since.

The research region is a warm temperate semi-arid and semi-humid type, with much rain in spring and summer and drought in autumn. The average annual temperature is 11.2 °C. The average annual precipitation in the Qilihai region is 620 mm, which is influenced by the monsoon and generally concentrated in July, August and September, accounting for 80% of the annual precipitation. The actual water storage in the region is 3 million cubic meters, and the soil is saline all year round.

The rich plant resources in the wetland park have the important roles of water conservation, water purification, flood and drought prevention, climate regulation, and biodiversity maintenance, which are of great significance in maintaining the ecological balance of Tianjin, realizing the harmony between man and nature and promoting the sustainable development of regional economy and society. At the same time, the functional value of the wetland habitat of Qilihai wetland is very important, with more than 180 kinds of rare and protected birds, which is of great significance for the study of land and sea changes and the coastal wetland ecosystem [18,19]. In addition, the precious wetland resources of the Qilihai is also a unique tourism and educational resource [20]. Therefore, this paper chooses the wetland as the case study. In 2009, in order to make the protected area more in line with the relevant standards of the national nature reserve and to protect the protected objects more effectively, according to the Regulations of the People’s Republic of China on Nature Reserves, the Regulations on the Adjustment of the Scope and Functional Areas of State-level Nature Reserves, and on the Adjustment and Change of Names of State Nature Reserves adjusted the scope of this reserve from the original total area of 980.606 km^2^ to a total area of 359.13 km^2^. Considering the consistency completeness of the study, the adjusted scope was selected for the years 2008 to 2017. The geographical location of Qilihai Wetland is shown in Figure 1.

### 2.2. Materials

(1)Data source

The remote sensing data used in the study on Qilihai Wetland was obtained from USGS (http://earthexplorer.usgs.gov) and Geospatial Data Cloud (http://www.gscloud.cn/search). The study selected remote sensing images data in 2008, 2011, 2014, and 2017. Among them, 2008 and 2011 are Landsat5 TM image data, and 2014 and 2017 are Landsat8 OLI image data. In addition, this research also obtained from the Ministry of Natural Resources of the People’s Republic of China (http http://www.mnr.gov.cn/) and the People’s Government of Ninghe District, Tianjin (http://www.tjnh.gov.cn/) obtained the administrative division map and land use status of Tianjin and Ninghe District The picture is for reference. This is according to the Report of the Comprehensive Scientific Investigation of Tianjin Ancient Coastal and Wetland National Nature Reserve compiled by the Tianjin Water Transport Engineering Research Institute of the Ministry of Transport and the Planning of Tianjin Ancient Coastal and Wetland National Nature Reserve formulated by the Tianjin Forestry Bureau [21,22]. According to the plan of Tianjin Ancient Coastal Wetland National Nature Reserve, the quality of the natural environment, the diversity of animals and plants, the socio-economic status and the operating status. The reserves have been mastered. This is combined with local field surveys, ground monitoring and analysis data, and meteorological and hydrological data.

(2)Data processing

Firstly, the satellite images were processed by ENVI 15.3 software (Harris Geospatial, Broomfield, CO, USA) for radiation correction, atmospheric correction, data fusion, and image enhancement. After pre-processing the images, supervised classification was performed using the maximum likelihood method. As a result, the satellite images were classified into species land-use categories: agricultural land, built-up land, water bodies, reeds, and swamps. Combined with the spatial analysis function of MapGIS, the analysis and interpretation of the land use patterns of the Qilihai wetland is performed to dynamically analyze the results of the classification.

### 2.3. Research Methodology

#### 2.3.1. Assessment Method

Ecosystem service value assessment methods include physical quality assessment, energy value analysis, and value quantity assessment [23]. The assessment methods differ for different areas and types of wetlands. The selection of wetland benefits should be made for the type with the most prominent benefits, while the selection of assessment methods should be based on their feasibility and operability [24]. In contrast, the results of the value quantity evaluation method are expressed in the form of monetary values, which can visually show the magnitude of the value of natural ecosystem services and help to attract the attention of the government and the public [25]. Secondly, it is easy to compare the value of different ecosystem services in an ecosystem to promote the improvement of management measures for natural ecosystems and sustainable development of ecosystems. Many of the estimation methods used in quantitative valuation methods are derived from ecological economics, environmental economics, and resource economics, including marketing value method, opportunity cost method, defensive expenditures, recovery cost usage, shadow engineering method, travel cost method, correspondence inquiry method, and conditional value method [25,26,27,28]. In general, methods for assessing the value of ecosystem services can be divided into three main categories: substitutable marketing value method, simulated marketing value method and direct marketing value method.

(1)Substitutable marketing value method

Johnston [29] et al., introduced the concept and method of benefit transfer to assess the ecological and economic value of fish habitat restoration. Xu [30] used the travel cost method to evaluate wetland resources in different cities such as Beijing, Hangzhou and Wuhan, and found that the “product function” of urban wetlands had been largely lost, and the “regulation function” was gradually declining. There is an increasing emphasis on “cultural functions”, and this estimation method is considered to be well suited to assessing cultural functions. The alternative market technique method can be used to derive the value of the ecological environment through side-by-side comparative analysis, but is more subjective, heavily influenced by other factors, and less credible.

(2)Simulated marketing value method

This method, which mainly includes conditional value estimation, simulates the market by asking questions to investigate the willingness to pay and net willingness to pay of groups to study the ecosystem goods and services of the region. The idea of the conditional value approach was first proposed by Ciriacy-Wantrup in 1947 [31], and Loomis [32] and others on the restoration of the ecological service functions of the Platte River Basin Willingness to pay study; Grazhdani [33] used the conditional value approach to assess the economic value of restoration of damaged river ecosystem services in the transboundary Buna region of Albania. The simulated market approach can evaluate the economic supply value of ecological service functions and is suitable for the assessment of resource values such as cultural heritage sites, animal habitats and species that are not of high utility value. In the early 21st century, the conditional value approach was introduced to assess resource values and is the most used simulated market approach in China [34]. The domestic scholar Xue [35] used the travel cost method and conditional value method to assess the tourism value of Changbaishan nature reserve and found that the tourism value of biodiversity in Changbaishan reserve was 432.05 million yuan in 1996. Currently, this method in China is mostly in the form of face-to-face and household interviews and questionnaires, which requires the surveyed users to honestly state their willingness to pay, and the credibility and reliability need to be improved [28].

(3)Direct marketing value method

The direct market approach, which includes market value, opportunity cost and shadow engineering approaches, treats the quality of the environment as a factor of production that can be priced in the market, and then evaluates whether improving the environment will bring benefits. Stephen [36] used the direct market approach to assess the value of wildlife and the environment in 1984, which was one of the earlier research cases. In China, different scholars have also made more use of the direct market approach. Ninan et al. [37] used the shadow engineering method to estimate the water-supply function of a protected forest ecosystem in India. Zhao [38] used the market price method to evaluate the service index system of a grassland in Qinghai province to provide a basis for policy formulation for local ecological resource management and ecological compensation. The direct market method tends to be more objective and comprehensive and can credibly estimate difficult-to-estimate ecological values, but the data requirements are high, and if the shadow engineering method is used, it requires alternative engineering non-uniqueness.

As each method of assessing the functional value of ecosystem services has its own strengths and weaknesses, a single method is often not used in practice, but rather a combination of different methods for different categories of value based on the ecosystem service classification. There are many ways to classify ecosystem services, including functional, organizational and descriptive classifications [8,39,40,41], and functional classification is the main classification for ecosystem service evaluation. With reference to the Millennium Ecosystem Assessment, the ecosystem services of the Qilihai Wetland in Tianjin were further classified. The ecosystem services of the Qilihai Wetland were divided into four types: supply services, regulation services, support services, and cultural services. By summarizing the existing literature and comparing the applicability of each method, the assessment method for each type was determined. The direct market approach is mainly used for the regulatory functions of the ecosystem; the market value approach is mainly used for production functions such as agricultural and aquatic products; and the conditional value and market value approaches are mainly used for cultural functions. The ecosystem services of the Qilihai wetland were subdivided into eight functional types, taking into account the characteristics, structure and ecological processes of the study area. Among them, supply services include material resource supply of one second level; regulation services include climate regulation, flood storage and water purification of three second level; cultural services mainly include tourism, scientific research and education of two second level; and support services include soil conservation and maintenance of biodiversity of two second level.

#### 2.3.2. Calculation Method

(1)Supply services

According to the field study, and combined with remote sensing image interpretation, the main land use types in the Qilihai wetland are reeds, water bodies and agricultural land. Reeds are the dominant species, which cover 60–80% of the whole wetland. The reed can be used for paper making and textile, and the value of raw material production is high. At the same time, the roots of reeds can absorb nitrogen and phosphorus in the water during the growth process and have many functions such as climate regulation, water purification and maintaining biodiversity. The value of reed is evaluated using the direct market approach to calculate only its direct value. The value of reed after processing is not included in this calculation because of other costs involved in post-processing. Dry reed yielded 6000–7500 kg/hm^2^ per unit, which was calculated based on 6500 kg/hm^2^ per unit at a market price of 300 Yuan/t [42]. The harvestable area was calculated as 50% of the total production area. The production of silver fish, purple crab and other foodstuffs in the aquatic product base is huge and can bring great economic value. According to the data, the Qilihai Wetland has 3000 hm^2^ of available breeding water surface, fish and crab breeding in 1700 hm^2^ and 1000 hm^2^, respectively, with an annual output of 1.2 × 106 kg and 5 × 105 kg. The main fish species is calculated according to its average market price of 16 yuan/kg. In addition, the grain production can be estimated according to the area ratio according to the Tianjin Statistical Yearbook. There are about 10,000 hm^2^ of cultivated land in the protected area and about 30,000 hm of rice cultivation, with a total output of about 2 × 107 kg. The grain price is calculated according to the market price of 6 yuan/kg.

(2)Atmospheric regulation services

Atmospheric regulation services are one of the important functions of wetland ecosystems and can effectively improve air quality. With the continuous development of industry, the concentration of CO_2_ in the atmosphere is increasing, leading to global warming, which has caused widespread concern and anxiety. Through the photosynthesis of wetland plants, they continuously absorb CO_2_ and release O_2_, which plays an irreplaceable role in maintaining the dynamic balance of CO_2_ and O_2_ in the atmosphere and reducing the urban heat island effect. According to the plant photosynthesis equation: 6CO_2_ + 12H_2_O = 6 (CH_2_O) + 6O_2_ + 6H_2_O, plants need to absorb 1.63 g of CO and release 1.19 g of O_2_ for every 1 g of dry matter they produce, the value of CO_2_ absorption and O_2_ release in wetlands can be calculated according to this equation. The average NPP of Tianjin is 321.90 gC/a m^3^ [43,44], which is based on the cost of afforestation in China and the industrial oxygen generation standard. The value of carbon sequestration is taken as the arithmetic mean of RMB 625.45/t using the Chinese afforestation cost of 260.90 Yuan/tC (constant 1990 price) and the internationally common Swedish carbon tax rate of 150 USD/tC (RMB/USD exchange rate of 6.60 Yuan at the end of 2017), and the industrial oxygen generation price is 0.4 Yuan/kg.

(3)Flood storage services

Wetland soils have strong hydrological functions and are natural “sponges” for water storage and flood control. Its unique ecological structure can not only absorb a large amount of precipitation and transit water, but also slow down evaporation and transpiration, and maintain the total amount of water resources in the region in a long-term stable manner. During the high water period, the wetland can prevent flooding through its water storage, and excessive river water and surface runoff from upstream and surrounding areas can be stored directly here; in times of drought and water shortage, water can be transferred from the wetland’s storage to supply water for industrial and agricultural production. Thus, the wetland not only plays the role of flood and disaster prevention, but also alleviates the shortage of freshwater resources to a certain extent. According to the shadow engineering method, the hydrological functional value is estimated using the storage capacity cost and the maximum amount of flood water that can be stored in the wetland. According to the national reservoir construction investment from 1997 to 2009, taking into account the price increase index and considering the actual situation of the Qilihai area, the average construction price per unit of reservoir capacity was determined to be 6.1 Yuan/m^3^.

(4)Water purification services

Wetlands have a powerful water purification function and provide a natural space for humans to deal with pollution. Wetland ecosystems use physical, chemical, and biological synergies to degrade and purify pollutants in water through filtration and adsorption by soil, absorption by plants, and degradation by microorganisms. The pollutants can also be completely removed from the system through regular harvesting of aquatic plants [45]. The main wastewater purified at the Qilihai wetland includes two parts, industrial and urban wastewater from the upstream towns, and agricultural wastewater from the wetland. The value of this function can be assessed by the shadow engineering method, i.e., assuming the construction of a wastewater treatment plant with the aforementioned capacity to purify water quality in the wetland; the construction cost and its operating cost can represent the value of the wetland water quality function.

(5)Biodiversity generation and maintenance services

The Qilihai wetland is one of the richest and most distinctive areas in terms of biodiversity. Its large areas of reed swamps, shallow banks and ponds provide a natural derivative site for wildlife. The special habitat of the wetland provides a rich source of food and a good place for wading birds and migratory birds to nest and hide from enemies. There are 41 families and 153 kinds of wild plants in the wetland, accounting for 93% of the plants in the wetland of Tianjin Binhai. There are a large number of ornamental plants in the wetland. There are 108 species of traveling birds in the wetland, accounting for 60.02% of the total number. The numerous wild animal and plant resources play an important role in maintaining the ecological balance of nature. The biodiversity value is obtained from the ecological service value of biodiversity per unit area and the total area of the wetland. This is done by using the annual ecological benefit of US$439/hm^2^ (RMB/USD exchange rate of 6.60 Yuan at the end of 2017) from the results of the value of bioshelter services per unit area of coastal zone salt marsh wetlands calculated by Costanza et al. and combining the value of biodiversity maintenance services per unit area of wetlands in China of 2212.2 Yuan/(hm^2^·a) for the Biodiversity value assessment [4,46]. The arithmetic mean of the above two calculations, 2554.8 yuan/(hm^2^·a), will be used as the average value of biodiversity maintenance services per unit area of the wetland [47].

(6)Conservation of soil nutrients and reduction of waste soil

Today, about 20% of the world’s land is degraded as a result of human activities (Oldeman et al., 1991). In addition to their role in the water cycle, the ecological service functions of soils include, at a minimum, the provision of sites for plant growth and development; the conservation and provision of nutrients to plants; the key role of soils in the reduction of organic matter; the rendering harmless of many potential human pathogens (wastes); and the degradation of organic matter and the cycling of nutrients. These functions can be summarized as soil nutrient retention and waste reduction. Wetlands are affected by both deep-water and terrestrial systems, with terrestrial system materials entering the wetland system during hydrological processes, often contributing to the development of unique soil conditions downstream of the wetland. The USLE model is used to simulate the field measurements to obtain the corresponding data, and rainfall, slope length, vegetation and soil are used to evaluate the strength of the ecosystem soil conservation function, and the cost substitution method is used to obtain the value of the wetland’s soil nutrient conservation function and waste reduction function. The soil loss equation and its factors are as follows. The soil loss equation is A = R*K*L*S*C*P. A is the annual average soil loss per unit area, R is the rainfall erosivity factor, K is the soil erodibility factor, L is the slope length factor, S is the slope factor, C is the vegetation coverage and management factor, and P is the soil and water conservation measures factor. The use value of wetlands to reduce soil nutrient loss is replaced by the value of nitrogen, phosphorus, and potassium in the soil. Only about 40% of the nitrogen, phosphorus, and potassium in agricultural fertilizers is available to crops each year; the remaining 60% is lost in various forms. Agricultural fertilizers in the area around the Qilihai Wetland are mainly nitrogen and phosphorus fertilizers, and the amount of potassium fertilizer is negligible, so potassium fertilizer is ignored in the calculation. The nitrogen and phosphorus contents of the soil in the marshes of the Qilihai wetland in Tianjin were 1.6 g/kg and 0.5 g/kg, respectively [48,49]. The prices of fertilizers were based on the import prices of urea, nitrogen–phosphorus–potassium compound fertilizer, and diammonium phosphate from China Statistical Yearbook. The average annual income from agriculture was obtained from the Tianjin Municipal Statistical Yearbook and Ninghe District National Economic and Social Development Statistical Bulletin.

(7)Tourism service

The Qilihai is one of the world’s three most famous ancient coasts, and is the first and only state-level nature reserve in China where an ancient coast and a wetland are co-located, with major natural relics such as an oyster beach (reef), stripe herringbone and moose horn. There are no other oyster beaches in China, and they are also rare in the world. Nowadays, people advocate a return to simplicity and nature, making the Qilihai Wetland a hotspot for ecotourism. Qilihai is fresh and clean, known as the Tianjin-Jingtang area “the largest natural oxygen bar”, Tianjin’s “green lungs”. Since 2011, the local area built a National Wetland Park in the core area and buffer zone, constructing a large amount of tourism infrastructure and carrying out tourism business activities to destroy and change the original wetland habitat of the protected area. In 2014, the entire area of the protected area was destroyed. The municipal government has included the Tianjin City Ecological Land Protection Red Line area for permanent protection, and tourism activities are not allowed. Subsequently, the Qilihai National Wetland Park shut down and gradually dismantled the related facilities, and is currently in a closed state. Therefore, this thesis only counts the tourism value in 2014. According to the survey, during the opening period, the Wetland Park received 300,000 to 400,000 visitors a year.

(8)Research and education services

The rare natural relics, rich resources and unique environment are of high scientific research value. It is of great importance that these typical natural monuments should be made available to the public, especially to primary and secondary school students, for the purpose of popularizing science education on the ecological environment and nature conservation. Wetland ecosystems have unique geological and hydrological conditions and abundant natural resources, which are of high educational and cultural value and can be used to educate the public, especially young people, about ecology. At the same time, the Qilihai Wetland’s unique structure and functional plant and animal communities provide scholars in many fields with a good natural base and an ideal place for scientific experiments. Therefore, the Qilihai wetland has great scientific and cultural value in the study of land and sea changes, biodiversity research, science education, and ecological environment education. Only the area of marshes, reeds, and water bodies were included in the calculation process. The average scientific research value of wetland ecosystems per unit area in China, 382 yuan/hm^2^, and the arithmetic mean value of Constanza et al., for the scientific research and educational function value of global wetland ecosystems, 861 USD/hm^2^ (RMB/USD exchange rate of 6.60 yuan at the end of 2017) and 3032.3 yuan/hm^2^, were selected to calculate the scientific research value of wetlands [2,50]

## 3. Result and Discuss

### 3.1. Land Use Change

Based on Landsat satellite images, this paper draws a land use map from 2008 to 2017 (see Figure 2 for details) and compares the area and change trends of different land use types (see Table 1 for details). Taking the 2008 land use change map as an example, farmland and reed are the largest land use types in the study area, accounting for 26% and 37% of the entire wetland area respectively. In contrast, marshes accounted for only 10% of the study area. It should be mentioned that the wetland water body covers an area of 12,399,543 m^2^, which is 14% of the study area. Farmland expanded rapidly in 2011 and then showed a downward trend in 2014. However, the changing trend of the water body area is just the opposite, showing a downward trend in 2011, and then gradually increasing after 2013. The expansion of water bodies can be attributed to the conversion of farmland. In addition, the area of reeds and swamps has continued to decline in the past 10 years, which has greatly affected the regulation and support functions of the Qilihai Wetland.

### 3.2. Service Value Analysis

According to the ecosystem service functions of Qilihai Wetland and its evaluation method provided in Table 2, the service value of different ecosystem services from 2008 to 2017 can be calculated, as shown in Table 3. Figure 3, Figure 4 and Figure 5 shows the change of ecological service value of Qilihai Wetland from 2008 to 2017

#### 3.2.1. Overall Trend Analysis

By evaluating the value of the above four types of service functions of the Qilihai Wetland in different years, it can be concluded that the overall ecological service value has shown an increasing trend. In 2008, the total ecological service value of the Qilihai Wetland was 317 million yuan, with an average value of 35,791.6 Yuan/hm^2^. In 2011, the total ecological service value of the region was 299 million yuan, with an average value of 33,759.3 yuan/hm^2^; in 2014, the total ecological service function value of the region was 283 million yuan, with an average value of 31,952.8 yuan/hm^2^; the value of regional ecological service functions is 321 million yuan, with an average value of 36,243.3/hm^2^. The overall service function value of Qilihai Wetland shows a trend of first decline and then rise. Due to improper planning, the development of tourism in the reserve and even the core area since 2010 has greatly affected its ecological conditions. This is also consistent with the results of Jalil Badamfirooz (2019). They reported that between 1975 and 2013, as the construction area and arable land expanded, pasture and woodland in the Anzhali Wetland Basin were lost. Although the ES of the supply function has increased in this case, the ES of erosion control, nutrient recycling, climate treatment, and water treatment has dropped significantly. This is the main contributor to the downward trend in the total ES of the landscape. In 2015, the Qilihai Management Committee gradually began to demolish illegal development projects and began rectification and restoration of wetlands. Different ecological protection measures have greatly changed the land use structure of the study area; further, this significantly changed the ecological conditions of the study area and changed the value of regional ecosystem services.

According to the magnitude of value, the order of the value of the ES of Qilihai Wetland in 2017 is supply service > regulation service > support service > cultural service. Among them, the value of aquatic products and plant products is in a dominant position, accounting for 61.54% of the total value. In 2014, the order of value of ES of Qilihai Wetland was supply service > regulation service > cultural service > support service. Among them, the value of aquatic products, the value of plant products, the value of tourism, and the reduction of waste soil four types of ES accounted for 68.57% of the total value. In 2011, the value of ES of Qilihai Wetland was ranked in the order of support service > supply service > regulation service > cultural service. Among them, the value of aquatic products, the value of plant products, and the reduction of waste soil accounted for 65% of the total value. In 2008, the order of ES value was supply service > support service > regulation service > cultural service. Among them, the value of aquatic products, plant products and reduction of waste soil accounted for 65% of the total value. The regulation service value has increased significantly in the past 10 years. The supply service value and support service value accounted for a relatively high and stable state, while the cultural service value has not changed much in the past 10 years and has a relatively low share. This shows that the Qilihai Wetland has a relatively large proportion of wetland development and utilization, but the utilization of ecological service functions is still relatively single and overly dependent on the direct supply of wetland resources.

#### 3.2.2. Supply Service Change Analysis

In the last decade, the material supply service has been the core function of the Qilihai Wetland and the most direct kind of ES. This is due to the fact that the Qilihai Wetland contains large areas of reed land, arable land and farmed fish ponds, and these products, such as rice, fish, reeds, and pasture, have direct access to markets and create value. From the chart, it can be seen that the proportion of material product value is increasing year by year, which shows that people’s use of wetland ecosystems is still at a more elementary stage, and the productive value is greater than the ecological service value. The proportion of fish supply service value is increasing, from a low of 14.68 in 2011 to a high of 43.73% in 2017. Prior to 2011, villagers in and around the reserve polder natural wetlands for farming and cultivation in order to earn a living, resulting in a decrease in the area of natural wetlands and their conversion to artificial wetlands or cultivated land. Since 2011, a series of regulations on wetland protection have emerged as people have become more aware of the ecological value of the wetland and have attached greater importance to it. The government has introduced measures to permanently protect the Qilihai Wetland, related artificial facilities have been removed, the occupation of natural wetlands has been curbed, and the water body area has increased significantly, by 128% as of 2017. On the other hand, the COD, TN, and TP have all been at severely polluted levels. This has an important connection with the pattern of the Qilihai Wetland, which is mainly aquaculture. The Qilihai Wetland is not a source of drinking water, but a body of aquaculture water. Aquatic plants can absorb nutrients such as nitrogen and phosphorus in the water body, and absorb them into their own tissue structure. The eutrophication of aquaculture water is beneficial to improve the production and quality of aquatic products and reeds; while planting reeds can effectively absorb nitrogen, phosphorus and other eutrophication substances in the water body, and play a role in purifying water quality and improving water transparency. This is conducive to the development, utilization and protection of Qilihai Wetland [51,52].

#### 3.2.3. Regulation Service Change Analysis

In recent years, the regulation service of Qilihai Wetland (e.g., climate regulation function, water purification function, and flood storage function) has maintained a steady trend, but the percentage of this service is low, ranging from 19.65% to 21.97%. This phenomenon is related to the changes in water and reed area in the land use type of the Qilihai Wetland [47]. From an ecological point of view, reeds are vegetation types with multiple ecological functions, which have important roles in mitigating soil politics, maintaining soil and water, and regulating air humidity. From 2011 to 2017, the reed wetlands in the core area decreased by 32.8% and the water body area increased by nearly 128% as reed ponds were contracted in various ways for fish, shrimp, and crab farming. The reclamation and destruction of reed wetlands has resulted in a significant reduction in reed area, reducing the regulation service value that reed wetlands bring to the water body and increasing the regulatory value that reed wetlands bring to the water body. The water body area of the Qilihai Wetland is mentioned in the Tianjin Wetland Nature Reserve Plan to carry out protection and restoration work to give full play to the ecosystem service functions of wetlands such as purifying the atmosphere, improving the environment, regulating the climate, improving the soil, and conserving water. By 2025, the ecological quality of the wetlands in the reserve will be comprehensively improved, 40 km^2^ of reed restoration will be completed, and a scientific reed conservation system will be built. These measures will reduce villagers’ damage to reed lands and greatly improve the regulatory function of the Qilihai Wetland.

#### 3.2.4. Cultural Service Change Analysis

Recreational services are another important ecosystem service of the water body, the two main items of tourism and scientific research. In the last decade of ecological value assessment, the value of research and education has changed little, fluctuating between 4.31% and 5.17%. At the end of 2010, the West Sea began to develop the Qilihai Wetland Park tourism project, with only 2014 tourism revenue in the selected years, 3.6 × 108 yuan. Without properly designed tour routes and areas, tourists can enter the core area of the reserve by boat or on foot. The influx of tourists into the core area generates a lot of litter and even damages the vegetative landscape. This has had a negative impact on the function and condition of the wetland. Since it is illegal to develop and operate tourism in the core and buffer areas of the reserve, since late May 2017, Ninghe District has begun to remove tourism facilities. The core area was closed for management and replanted with various types of ecological grasses to restore the natural original appearance and ES of the wetland. While this measure minimized the damage to the wetland caused by human actions, it did not make full use of the reserve’s unique tourism resources. During the “Twelfth Five-Year Plan” period, the Ancient Coastal and Wetland National Nature Reserve has completed nearly 30 key scientific research projects at the national and provincial and ministerial levels, the “Tianjin Binhai Area Ancient Coastal and Wetland Ecological Restoration Capacity Building and Mechanisms Research”, and a series of science and technology and the sea. The project has improved the quality of the wetland water environment and added a demonstration area for integrated aquatic bioremediation technology in the Qilihai Wetland. In addition to the natural landscape, the Qilihai Wetland has a profound cultural and historical heritage, as well as a wealth of folklore, humanities, history, and other tourism resources. In accordance with the Regulations of the People’s Republic of China on Nature Reserves, the management agency of the reserve proposes to “carry out ecological tourism”. On the premise of scientific protection and in accordance with the requirements of the Tianjin Tourism Development Plan, the construction of ecotourism is re-planned. Outside the core area and buffer zone of the Qilihai, improve the ecological environment of the wetland, build an “ecological restoration zone of water system cultured wetland”, and at the same time support ecotourism projects such as bird watching, and carry out ecological breeding to produce fish, shrimp, crab and other products of the Qilihai Wetland. The surrounding buildings will be built as a folklore experience area, to develop the rich folklore, humanities, history and other tourism resources in the reserve, reflecting the humanistic heritage. It is expected that after the implementation of the tourism plan, without affecting the service value of other systems, the cultural service value of the wetland will be greatly enhanced.

#### 3.2.5. Support Service Change Analysis

The Qilihai Wetland Reserve is rich in natural resources, with an intact wetland ecosystem and high biodiversity. Among them, the shell dike and oyster reef belong to the underground protection object. Wild animal and plant resources are basically distributed in the oyster reefs, the core area and the buffer zone of the Qilihai Wetland. The ecosystem is mainly maintained because the wetland provides a rich source of food for birds and a good place for them to nest and hide from enemies. The waters and shallow areas in the protected area are large, which attracts more rare birds and plays an important role in the conservation of rare birds. The biodiversity value changed little in 2008, 2011, 2014, and 2017, but there was a declining trend of 13.8 million yuan, 12.08 million yuan, 11.82 million yuan, and 11.64 million yuan, respectively. The main reason is the continuous human interference and destruction since the end of the last century, theft and dredging of shell dikes and oyster reefs occurred. In addition, due to historical reasons, the vast majority of the reserve land is collectively owned. There are still 25,000 people living in five villages in the buffer zone, as well as a large amount of farmland and farming water. The conversion of wetlands into drylands and the abuse of herbicides by villagers have significantly affected the species and quantity of wild plants, causing the structure and diversity of plants in the reserve to change in an unfavorable direction, which has a great impact on the protection of rare species and the increase of ecological diversity value in the reserve. The Tianjin Wetland Ecological Compensation Measures (Trial) proposes to implement an ecological migration project and complete the relocation of ecological migrants in the buffer zone by 2020. The ecological replenishment works and implementation of ecological restoration works will be implemented simultaneously. It is expected that after the completion of the plan, the wetland can be effectively protected, the biodiversity of the wetland ecosystem will not be reduced and the ecosystem will not be degraded.

In the evaluation of ecological value in the past 10 years, the function of soil nutrient conservation and the function of reducing waste soil have declined significantly. The value of the 4 years in 2008, 2011, 2014, and 2017 were 95 million yuan, 104 million yuan, 32 million yuan, and 35 million yuan, respectively. The reasons for this are mainly agricultural sources, pastoralism, and fishing point source pollution. The land types with supporting services are water bodies, farmland and reeds. Farmland and arable land discharge wastewater with pesticide and fertilizer residues, bringing excessive load to the water bodies in the protected area. Secondly, the unorganized and untreated discharges of domestic wastewater from nearby villagers also bring excessive load to the water bodies, leading to a precipitous decline in the value of support service in the past decade. In addition, the development of tourism in wetland and the activities of foreign tourists have also brought environmental pollution and damage to the reserve. Natural wetlands such as marshes and reed areas have been converted into cash crop cultivation sites, which has damaged the wetland environment and reduced the environmental capacity, resulting in a steady decline in the value of the Huixian Wetland’s support service. However, after tourism withdrew from the Qilihai Wetland in 2014, the ES of the wetlands rebounded from 32 million to 35 million yuan, and it is expected that after a period of restoration, the value of the ecosystem support service will increase rapidly.

#### 3.2.6. Contrast with Huixian Wetland

The results of this study are compared with the results of various services of Guilin Huixian Wetland (See Table 4 for details). It can be seen from the Table 4 that the service value per unit area of Huixian wetland is 7.24 yuan/(m^2^·a). The supply service value is 0.76 Yuan/(m^2^·a), the regulation service value is 2.23 Yuan/(m^2^·a), the cultural service value is 1.33 Yuan/(m^2^·a), and the support service value is 2.91 Yuan/(m^2^·a). Since the Qilihai Wetland has large areas of cultivated land, reedbeds and fish ponds, the supply function has been central. In contrast, because the area of farmland in the Huixian Wetland constantly fluctuates with policy changes and there are fewer fish products, its supply function is much smaller than that of the Qilihai Wetland. In terms of regulating service value, the value per unit area of the Wetland is much lower than that of the Huixian wetland. Most of the regulating services come from the reed area, but tourism and uncontrolled farming have caused extensive damage to the reed areas, reducing its water purification and climate regulation service. In terms of cultural service value, the Huixian Wetland successfully applied for the National Wetland Park to attract tourists, while the Qilihai Wetland Park was closed at that time, so there is a big difference between the two. In terms of the value of support services, both wetlands suffered from the destruction of local water bodies caused by the irregular discharge of wastewater, which reduced the environmental capacity and rapidly decreased the soil conservation function. In addition to this, the wetland has been damaged by visitors.

The reasons for the differences are: different types of wetlands, different geographical locations, and different core services. The Qilihai Wetland focuses on provisioning services, while the Huixian Wetland focuses on regulating and supporting services. Secondly, the indicators and assessment methods used by researchers to assess the different types of services differ. Currently, there is no uniform definition and assessment method for ecosystem service assessment. Therefore, future research on the value of wetland ecosystem services suggests the establishment of a unified assessment system to quantify their value.

## 4. Conclusions

The study is conducted to assess the impact of land use change on the ecosystem value of the Qilihai Wetland. The results of the study show that the ES of the Qilihai Wetland has an important economic value. The total values in 2008, 2011, 2014, and 2017 were 317, 299, 283, and 321 million yuan, respectively. Changes in human use, natural conditions and socio-economic changes will all lead to changes in the value of ES in the Qilihai wetland. Land use is the most direct intervention of human activities on nature, and it has an irreversible negative impact on the environment during changes [53]. During the period from 2008 to 2017, land use for agriculture and water bodies showed an upward trend, while other types of land use mostly showed a downward trend. Overall, the changes in the ES values of each land use type in the wetland were in the same direction as the changes in area, indicating that land use options are critical to the environmental quality of the wetland. The value of some ES of the Qilihai Wetland has been declining year by year. The disturbance of wetland ecosystems by human activities has affected not only the spatial extent and trophic structure of the ecosystems, but also the functions of hydrological regulation, climate regulation, water purification, and biodiversity. The impacts go beyond the threshold of the self-adaptive and regulatory capacity of wetland ecosystems.

This study can make policy makers aware of the importance of land-use change in increasing the value of ES and target land-use planning within the study area. The ES assessment will be used as a guide to implement reasonable and effective protection of the wetland and enhance the ecological restoration of the Qilihai Wetland. Therefore, it is recommended that the government conduct preliminary research and land use planning. The main reason for the reduction of wetland area is the degradation and shrinkage of the wetland due to the competition of paddy fields for the water supply to the wetland, which prevents a large amount of wetland from being reclaimed as agricultural land. It is also necessary to carry out rational ecotourism design. The economic benefits generated by the tourism industry can be reinvested as a fund into wetland ecological conservation. Consider appropriate zoning of the wetland, according to different landscape features, divided into different areas: ecological reserves, folk culture areas, rare animals, and plants landscape areas, etc. While visiting the wildlife, do not disturb the rest of the creatures, people do not damage to the environment. Allowing visitors to observe wildlife up close without destroying the values of the conservation area. In addition, the necessary mandatory controls need to be established to strictly regulate the discharge of sewage. The nature of a wetland park is such that it must have a certain size and few man-made structures. Therefore, it is important to prevent its destruction by uncontrolled development and human activities. For example, the conversion of wetlands into agricultural land and building land. Finally, post-monitoring and maintenance are recommended. They help to increase the value of support and regulation services, increase the sustainability of the wetland and maintain its long-term development. Future research on this wetland could consider incorporating the evaluation results into the management decision system to provide data support for food such as ecological compensation about the wetland, so that the ecosystem service value of the wetland can be put into practice.

## Figures and Tables

**Figure 1 ijerph-17-07108-f001:**
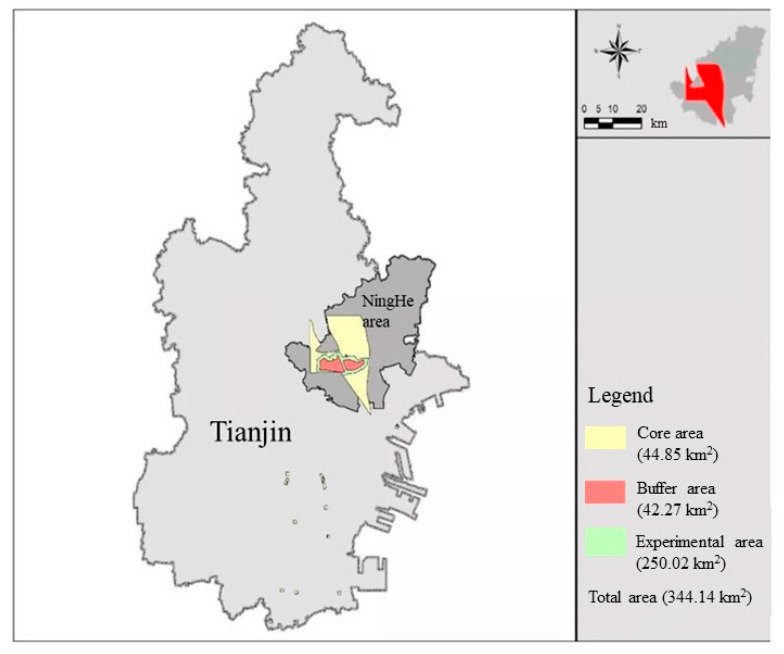
Geographical location of the Qilihai Wetland. (Source: The Qilihai Wetland Ecological Conservation and Restoration Plan (2017–2025)).

**Figure 2 ijerph-17-07108-f002:**
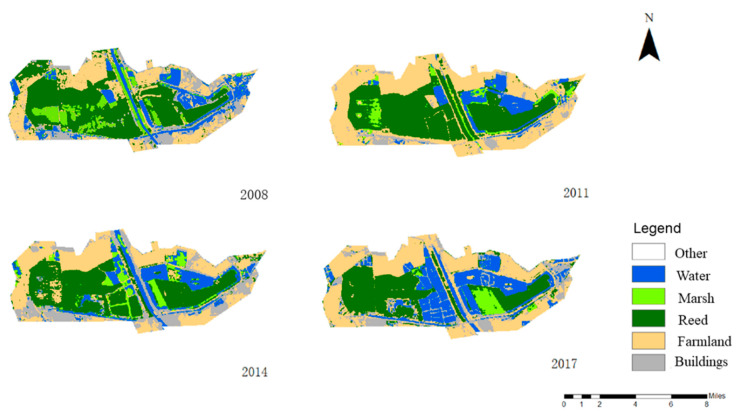
Land use change map of Qilihai Wetland from 2008 to 2017.

**Figure 3 ijerph-17-07108-f003:**
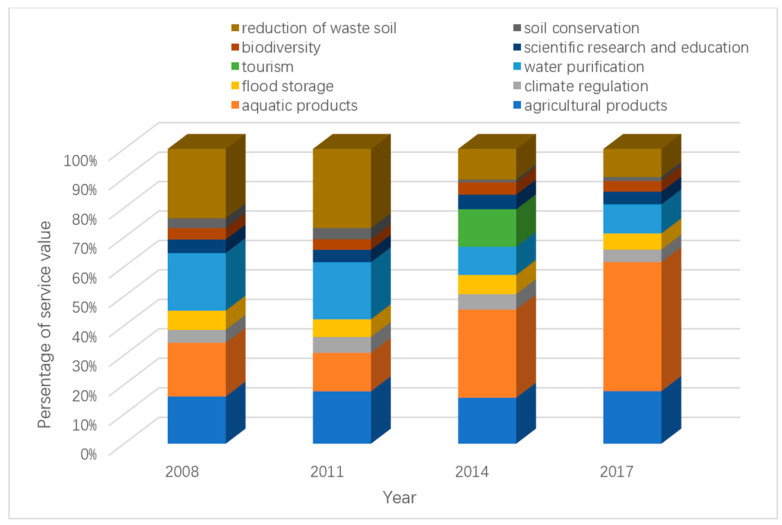
The percentage of subservice type from 2008–2017.

**Figure 4 ijerph-17-07108-f004:**
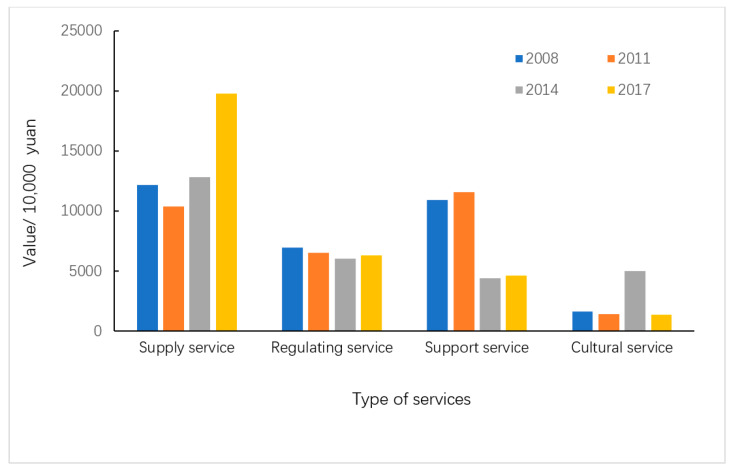
The value of the four main types of services from 2008–2017.

**Figure 5 ijerph-17-07108-f005:**
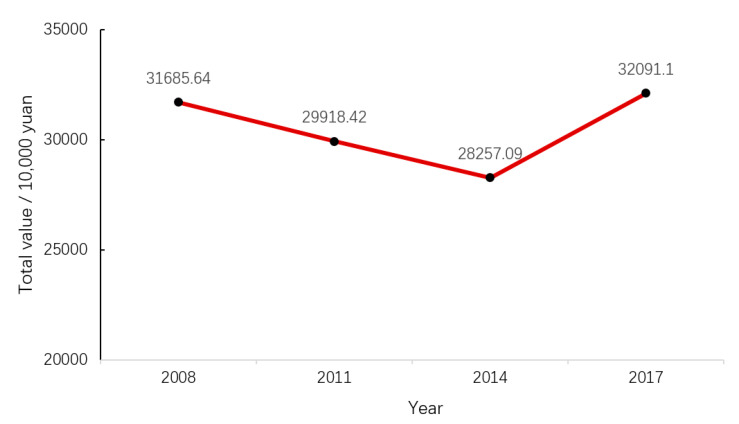
Trends in total value of ecosystem services from 2008–2017.

**Table 1 ijerph-17-07108-t001:** Land use area of Qilihai Wetland from 2008 to 2017.

Land Use	Year			
	2008 (m^2^)	2011 (m^2^)	2014 (m^2^)	2017 (m^2^)
**Farmland**	23,027,700	36,184,500	27,076,500	30,296,700
**Buildings**	11,513,850	5,086,800	15,191,100	12,701,700
**Water**	12,399,540	8,426,700	12,181,500	19,221,300
**Reed**	32,770,200	34,507,800	28,154,700	23,183,100
**Marsh**	8,856,810	4,362,300	5,964,300	3,165,300

**Table 2 ijerph-17-07108-t002:** ES of the Qilihai Wetland and the assessment methods.

Ecosystem Services	Value Types	Implication	Assessment Methods
**Supply Service**	Material production	Large area of reeds, mainly considering the value of reed production	Market value approach V = ∑ Si∙Yi∙Pi Si is the harvestable area of Group i substances; Yi is the yield per unit of Group i substances; Pi is the market price of Group i substances.
**Regulation Services**	Climate regulation	Plants absorb CO_2_ and release O_2_ through photosynthesis	Value of carbon sequestration: V = 1.63·R·Q·P_1_ + 1.19·Q·P_2_ R is the amount of carbon in CO_2_ i.e., 12/44; Q is the biomass of aquatic plants; P_1_ is the price of sequestered carbon; P_2_ is the price of oxygen.
	Flood storage	Storage of flood waters	Shadow engineering method: V = ∑ Vi·Pi Vi is the annual discharge of the different types of sewage; Pi represents the price of treatment of the different types of sewage.
	Water purification	Reeds grow to absorb nitrogen and phosphorus from water to purify sewage	Replacement cost method: V = A·H·K A is the area of the wetland capable of storing water; H is the variation of the water level in the wetland; K represents the cost of the reservoir per unit volume of water stored.
**Cultural Services**	Tourism	Provide places for leisure, tourism, entertainment, etc.	Travel cost method: V = Q·P A is the wetland area and P is the tourism revenue benefit per unit area.
	Scientific research and education	Sites for scientific research and youth education	Shadow engineering method: V = P∙A P is the research and education value per unit area of wetland; A represents wetland area.
**Support Services**	Maintaining biodiversity	Provide a good habitat for a variety of organisms	Outcome reference method: V = P∙A P is the value of biodiversity ecological services per unit area of wetland; A is the total local wetland area.
	Soil conservation		Shadow price method and opportunity cost method: V11 = ∑ Ac·Ni·Pi, V12 = Ac·B/(1000·d·ρ) V11 is the unit value of soil nutrient retention; Ac is the amount of soil retention; Ni is the net content of soil nitrogen, phosphorus, and potassium in the wetland; Pi is the price of fertilizer; V12 is the economic benefit of reducing abandoned land; Ac is the amount of soil retention; B is the average annual return from agriculture; ρ is the soil capacity and d is the soil thickness.

**Table 3 ijerph-17-07108-t003:** Ecosystem service value of Qilihai Wetland.

ES	Value Types	Year							
		2008		2011		2014		2017	
		Value (Ұ10,000)	Percentage (%)	Value (Ұ10,000)	Percentage (%)	Value (Ұ10,000 )	Percentage (%)	Value (Ұ10,000)	Percentage (%)
**Supply Service**	**Plant production**	5686.48	17.95	5988.71	20.02	4407.63	15.59	5716.43	17.81
	**Aquatic products**	6463.41	20.40	4392.52	14.68	8439.00	29.85	14,033.48	43.73
	**Subtotal**	12,149.89	38.35	10,381.23	34.70	12,846.63	45.43	19,749.91	61.54
**Regulation Services**	**Climate regulation**	1569.35	4.95	1821.79	6.09	1485.39	5.25	1374.94	4.28
	**Flood storage**	2306.93	7.28	2019.57	6.75	1835.81	6.49	1748.46	5.45
	**Water purification**	3086.58	9.74	2702.01	9.03	2714.31	9.60	3181.80	9.91
	**Subtotal**	6962.86	21.97	6543.37	21.87	6035.51	21.35	6305.20	19.65
**Cultural Services**	**Tourism**	0.00	0.00	0.00	0.00	360.00	12.73	0.00	0.00
	**Scientific research and education**	1638.24	5.17	1434.18	4.79	1403.97	4.97	1381.81	4.31
	**Subtotal**	1638.4	5.17	1434.18	4.79	5003.97	17.70	1381.81	4.31
**Support Services**	**Maintaining biodiversity**	1380.7	4.36	1208.34	4.04	1182.89	4.18	1164.21	3.63
	**Soil conservation**	1188.24	3.75	1287.35	4.30	261.72	0.93	428.18	1.33
	**Waste soil reduction**	8366.14	26.0	9063.96	30.30	2944.37	10.41	3061.77	9.54
	**Subtotal**	10,934.65	34.51	11,559.65	38.64	4388.98	15.52	4654.17	14.50
	**Total**	31,685.64	100.00	29,918.42	100.00	28,257.09	100.00	32,091.10	100.00

**Table 4 ijerph-17-07108-t004:** Comparison of the ES values between Huixian and Qilihai wetlands.

Huixian Wetland	Value (Yuan)	Size (m^2^)	Value/Size
**Supply service**	124,300,000	164,478,600	0.755721413
**Regulating service**	367,200,000	164,478,600	2.232509275
**Cultural service**	219,100,000	164,478,600	1.332088187
**Support service**	479,800,000	164,478,600	2.917096814
**Total**	1190,400,000	164,478,600	7.237415688
**Qilihai Wetland**	**Value (Yuan)**	**Size (m^2^)**	**Value/size**
**Supply service**	197,499,100	88,568,100	2.22991235
**Regulating service**	63,052,000	88,568,100	0.711904173
**Cultural service**	13,818,001	88,568,100	0.156015552
**Support service**	46,541,700	88,568,100	0.525490555
**Total**	320,911,000	88,568,100	3.623324877

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
