# Peer review of "Dynamic Evaluation of Ecological Service Function Value of Qilihai Wetland in Tianjin"

_ijerph, 2020, doi:10.3390/ijerph17197108_

Round 1

Reviewer 1 Report

  • Please, insert a numeric scale into the map of Figure 1.
  • Improve with more information par. 2.2 (Data Source and Data processing)

Author Response

Response to Reviewer 1 Comments

Dear Reviewer,
Thank your comments concerning our manuscript entitled “Dynamic Evaluation of Ecological Service Function Value of Qilihai Wetland in Tianjin” (ID: 933183). Those comments are all valuable and very helpful for revising and improving our paper, as well as the important guiding significance to our researches. We have studied comments carefully and have made correction which we hope meet with approval. Revised portion are marked in red in the paper. The main corrections in the paper and the responds to the your comments are as flowing:

Responds to the your comments:

  1. Response to comment: Please, insert a numeric scale into the map of Figure 1.
    Response 1:

Thank you for your advice. We have inserted a digital scale in the upper right corner of Picture 1, which reflects the ratio of the distance between the entire picture and the field. In addition, we have made further modifications to the pictures. The area of the core area, buffer zone, experimental area and total research area has been added to Picture 1, so that readers can read more easily and clearly. The specific drawing is as follows:

Line 127:(Since it cannot be displayed directly, please find the revised picture in the attachment)

Figure 1. Geographical location of the Qilihai Wetland

(Source: The Qilihai Wetland Ecological Conservation and Restoration Plan (2017-2025))

  1. Response to comment: Improve with more information par. 2.2 (Data Source and Data processing)

Response 2:

Thank you for your advice. Regarding your second advice, we have revised the paper. The main supplementary content is to increase the source of the image data website, the type of image data, the plan cited and the specific information of the report, etc.

The revised content of the article is as follows:

"The remote sensing data used in the study on Qilihai Wetland was obtained from USGS (http://earthexplorer.usgs.gov) and Geospatial Data Cloud (http://www.gscloud.cn/search). The study selected 2008 Remote sensing image data in 2011, 2014, and 2017. Among them, 2008 and 2011 are Landsat5 TM image data, and 2014 and 2017 are Landsat8 OLI image data. In addition, this research also obtained from the Ministry of Natural Resources of the People’s Republic of China (http http://www.mnr.gov.cn/) and the People’s Government of Ninghe District, Tianjin (http://www.tjnh.gov.cn/) obtained the administrative division map and land use status of Tianjin and Ninghe District The picture is for reference. According to the "Comprehensive Scientific Investigation Report of Tianjin Ancient Coastal Wetland National Nature Reserve" compiled by the Tianjin Water Transport Engineering Research Institute of the Ministry of Transport and the "Tianjin Ancient Coastal Wetland National Nature Reserve Planning" formulated by the Tianjin Forestry Bureau According to the plan of Tianjin Ancient Coastal Wetland National Nature Reserve, the quality of the natural environment, the diversity of animals and plants, the socio-economic status and the operating status. The reserves have been mastered. Combined with local field surveys, ground monitoring and analysis data and meteorological and hydrological data."

We tried our best to improve the manuscript and made some changes in the manuscript.  These changes will not influence the content and framework of the paper. And here we did not list the changes but they will marked in red in revised paper.

We appreciate for your warm work earnestly, and hope that the correction will meet with approval.

Once again, thank you very much for your comments and suggestions.

Reviewer 2 Report

The research question the essay tries to answer is relevant for the urbanization process in China: the impact of land use change on the ecosystem value; the essay considers the case of the Qilihai Wetland.

The essay is clearly presented, and it is an interesting story of what happened in the area in reference to the main policies. I suggest the paper offers more quantitative details about how and how much the different land use changes increased-decreased the ES and the related monetary value. In the conclusion some very interesting conclusion is given - lines 583-584, this point is worth exploring more: to what extent the land use change is irreversible? - but some other is rather well knows and generic - lines 599-602 - how is Qilihai Wetland specific?

I propose also to consider the following:

line 32: 4.4% of the global wetland: do you mean of the whole world?

line 33: in my country: in CHINA, I suppose

line 127: connected to the previous sentence?

line 136: who produced the Report and when? please explicit the source

line 342, 493: as the research mentions tourism several times, the research should take position: if tourism is not an acceptable activity in the area, the research should not consider it; if ecological tourism can coexist with the ES then the value of the ecological tourism should be considered, not that of 2014.

lines 364-365: please start a new page with the titles

line 447: the eutrophication causes severe alterations in wetlands, please explain further this point in the case study

line 543: the sentence is not complete.

line 543: some details about the Guilin Huixian wetland should be given such as dimension, activities, year of constitution; if the two cases are so different, as declared, why are they compared?

Author Response

Response to Reviewer 2 Comments

Dear Reviewer,
Thank your comments concerning our manuscript entitled “Dynamic Evaluation of Ecological Service Function Value of Qilihai Wetland in Tianjin” (ID: 933183). Those comments are all valuable and very helpful for revising and improving our paper, as well as the important guiding significance to our researches. We have studied comments carefully and have made correction which we hope meet with approval. Revised portion are marked in red in the paper. The main corrections in the paper and the responds to the your comments are as flowing:

Responds to the your comments:

  1. Response to comment: I suggest the paper offers more quantitative details about how and how much the different land use changes increased-decreased the ES and the related monetary value.

Response 1:

Thank you for your advice. Regarding quantitative research, we analyze the changes in land use based on the interpretation of the impact of satellite remote sensing in the selected years (ie, 2008, 2011, 2014, 2017). According to land use changes, use appropriate methods to account for changes in ecosystem service value caused by land use changes. Among them, the specific accounting analysis method is in the article 2.3.2. In addition, in 2.2 of the article, we verified and supplemented some relevant land use data sources in order to clarify the experimental methods.

"The remote sensing data used in the study on Qilihai Wetland was obtained from USGS (http://earthexplorer.usgs.gov) and Geospatial Data Cloud (http://www.gscloud.cn/search). The study selected 2008 Remote sensing image data in 2011, 2014, and 2017. Among them, 2008 and 2011 are Landsat5 TM image data, and 2014 and 2017 are Landsat8 OLI image data. In addition, this research also obtained from the Ministry of Natural Resources of the People’s Republic of China (http http://www.mnr.gov.cn/) and the People’s Government of Ninghe District, Tianjin (http://www.tjnh.gov.cn/) obtained the administrative division map and land use status of Tianjin and Ninghe District The picture is for reference. According to the "Comprehensive Scientific Investigation Report of Tianjin Ancient Coastal Wetland National Nature Reserve" compiled by the Tianjin Water Transport Engineering Research Institute of the Ministry of Transport and the "Tianjin Ancient Coastal Wetland National Nature Reserve Planning" formulated by the Tianjin Forestry Bureau According to the plan of Tianjin Ancient Coastal Wetland National Nature Reserve, the quality of the natural environment, the diversity of animals and plants, the socio-economic status and the operating status. The reserves have been mastered. Combined with local field surveys, ground monitoring and analysis data and meteorological and hydrological data."

  1. Response to comment: In the conclusion some very interesting conclusion is given - lines 583-584, this point is worth exploring more: to what extent the land use change is irreversible?

Response 2:

Thank you for your advice. We have added relevant grounds for the irreversibility of land use changes in the article. The revised content is as follows:

 "Land use is the most direct intervention of human activities on nature, and it has an irreversible negative impact on the environment during changes." (Chen Xing. Hydrological response to land use changes in watersheds under uncertainty: Taking Dianchi watershed Take an example[D]. Peking University, 2012.)

  1. Response to comment: lines 599-602 - how is Qilihai Wetland specific?

Response 3:

Thank you for your advice. We think that the particularity of Qilihai Wetland lies in:

(1) It is the only marine nature reserve in the country that does not involve modern coastlines, and it is also a national nature reserve. Tianjin is an important post for migratory bird migration in my country and even East Asia-Australia. The Qilihai Wetland is a characteristic area of ​​Tianjin’s biodiversity, with many rare birds and other precious animal and plant resources.

(2) Qilihai Wetland plays an important role in maintaining the ecological balance of Tianjin.

With the 19th National Congress of the Communist Party of China putting forward new requirements for the construction of ecological civilization and the new normal of biodiversity protection, the protection and restoration of the reserve is facing major opportunities and challenges. Studying the Qilihai Wetland and putting forward sustainable suggestions aims to achieve the three major benefits of ecology, society and economy by realizing the five major combinations of protection, scientific research, education, production, and tourism. This can help the reserve become a national ecological restoration model project, an international bird watching resort and an international ancient coast science education base.

  1. Response to comment:

line 32: 4.4% of the global wetland: do you mean of the whole world?

Response 4:

Thank you for your advice. Yes, it means that China's wetland area accounts for 5.58% of China's land area (ie wetland rate), and China's wetland area accounts for 4.4% of the global wetland area. The data is referenced from literature, and this information is also available in the government network of the State Forestry and Grassland Administration.

(http://www.forestry.gov.cn/main/4155/20140114/663608.html)(http://www.forestry.gov.cn/main/586/20200907/143713096119837.html)

The corresponding expression in the article has been modified to make the meaning of the article more accurate.

Line 32:  “The second national survey of wetland resources shows that China's wetland area is 53.62 million hectares, accounting for 5.58 percent of the country's total wetland area. And China's wetland area accounts for 4.4% of the global wetland area.”

  1. Response to comment: line 33: in my country: in CHINA, I suppose

Response 5:

We are very sorry for our incorrect writing that. Thank you for your advice!

We have revised the content of the article. Modified to "Compared with the results of the first survey ten years ago, the total area of wetlands in China under the same caliber has decreased by 3,396,300 hectares, a decrease rate of 8.82%."

  1. Response to comment: line 127: connected to the previous sentence?

Response 6:

Thank you for your advice. The article has been revised, and line 126 is revised to read "The geographical location of Qilihai Wetland is shown in Figure 1."

  1. Response to comment: line 136: who produced the Report and when? please explicit the source

Response 7:

Thank you for your advice. "Tianjin Ancient Coast and Wetland National Nature Reserve Comprehensive Scientific Investigation Report" was compiled by the Tianjin Ancient Coast and Wetland National Nature Reserve Management Office of Tianjin Water Transport Engineering Research Institute of the Ministry of Transport in December 2012; "Tianjin Ancient Coast "Planning for National Nature Reserves and Wetlands (2016-2025)" was compiled and completed by the Tianjin Ancient Coast and Wetland National Nature Reserve Master Planning Team in April 2018. We will make appropriate citation changes in the article.

  1. Response to comment: line 342, 493: as the research mentions tourism several times, the research should take position: if tourism is not an acceptable activity in the area, the research should not consider it; if ecological tourism can coexist with the ES then the value of the ecological tourism should be considered, not that of 2014.

Response 8:

Thank you for your advice. On the whole, ecotourism can coexist with ES. The state’s management and control requirements for nature reserves allow for tourism development in the experimental area of the reserve, but not in the core area and buffer zone.

However, in 2014, Qilihai Wetland Park had a tourism development phenomenon, and tourism was developed in inappropriate places. Although tourism value is generated, other service values are destroyed. Therefore, the tourist facilities were demolished after 2014. The reason why we only calculate the tourism value in 2014 and not other years. According to the actual existing behavior, the research area can only be developed and generate revenue. In the future, the research area still needs to use eco-tourism to improve the well-being of local residents. The position of this article is that we hope to establish a suitable method system to develop the research area and turn its ecological value into economic value. In addition, we hope to provide an economic foundation for higher levels of protection and ecosystem service functions through development

  1. Response to comment: lines 364-365: please start a new page with the titles

Response 9:

Thank you for your advice. We have made changes. Lines 362-363 of the article are new pages starting with the title.

  1. Response to comment: line 447: the eutrophication causes severe alterations in wetlands, please explain further this point in the case study

Response:

Thank you for your advice. We will put further explanations into the paper. Aquatic plants can absorb nutrients such as nitrogen and phosphorus in water bodies and assimilate them into their own tissue structure. (Zhao Xiang, Hui Feng, Lu Jianguo. Application of aquatic plants in eutrophic water treatment projects[J]. Modern Agricultural Science and Technology, 2013(04):237-238.). Proper planting and breeding can effectively absorb nitrogen, phosphorus and other eutrophication substances in the water body, play a role in purifying water quality, improving water body transparency, and improving the quality of aquatic products. (Qi Zhengliang, Bao Chengrong, Li Qian, et al. Ecological planting and breeding techniques of sand pond snakehead and river crab, green shrimp and reed rice[J]. Aquaculture, 2016(11):17-19.)

  1. Response to comment: line 543: the sentence is not complete.

Response:

Thank you for your advice. We modify the sentence of the article.

line 543: “The results of this study are compared with the results of various services of Guilin Huixian Wetland. It can be seen from the table below that the service value per unit area of Huixian wetland is 7.24 yuan/(m2·a). The supply service value is 0.76 Yuan/(m2·a), the regulation service value is 2.23 Yuan/(m2·a), the cultural service value is 1.33 Yuan/(m2·a), and the support service value is 2.91 Yuan/(m2·a).”

  1. Response to comment:

line 543: some details about the Guilin Huixian wetland should be given such as dimension, activities, year of constitution; if the two cases are so different, as declared, why are they compared?

Response:

Thank you for your advice. The reasons for comparing the two are as follows:

(1) Guilin Huixian Wetland and Qilihai Wetland are both wetlands approved by the State Forestry Administration to be included in the pilot construction of the National Wetland Park. An important reason for studying the Qilihai Wetland is also to provide sustainable development opinions for the subsequent construction of the National Wetland Park.

(2) The wetland ecosystems of the two places have been severely damaged, and both are related to the intensification of human activities in the surrounding villages. Both wetlands have been damaged and encroached, and the water area is gradually shrinking. Corresponding measures have been taken to maintain ecosystem services.

(3) The types of ecosystem services provided by the two are roughly the same, and both are rich in wetland resources, animal and plant resources, and tourism and cultural resources. We hope to compare and refer to Guilin Huixian Wetland, and make suggestions for the subsequent management and planning of Qilihai Wetland.

We tried our best to improve the manuscript and made some changes in the manuscript.  These changes will not influence the content and framework of the paper. And here we did not list the changes but they will marked in red in revised paper.

We appreciate for your warm work earnestly, and hope that the correction will meet with approval.

Once again, thank you very much for your comments and suggestions.

Reviewer 3 Report

The reviewed article is of very high quality when it comes to novelty, methods of research and conclusions. In reviewer’s opinion the study should be published due to its importance and showing to the reader usefulness of methods like physical quality assessment, energy value analysis, and value quantity assessment. Nevertheless, it should be stressed out that the reviver accepts whole part 2.2 (materials and methods), 3. (Result and discuss), 4. (Conclusion), but the introduction part should be done more precisely.

For example, authors begin their elaboration with the definition of wetlands – this definition is of course theoretical, in reviewer opinion authors should give some legal definition of the meaning (please see for example Shine, C., & De Klemm, C. (1999). Wetlands, water, and the law: using law to advance wetland conservation and wise use (No. 38). IUCN. Afterwards, authors should give legal definition of wetland (if there is any) that is appropriate for the Tianjin region (or maybe for PRC). This is of the essence because the reader needs to know if there were any obstacles in the research part of the study. Very often legal definitions makes it hard to understand if something can be even treated as a wetland or a lake or a sea (like for example: in legal terms Caspian Sea is a Sea, but in geographical and physical way is an Endorheic basin with legal status and regime – lake sui generis) – therefore a wetland with a just one simple legal act can be treated as a normal land – and opposite.

Authors writes about Sustainable Development Goals and quote appropriate literature – also in MDPI journals there are  a lot of possible refences to SDG’s when it comes to waters/lakes and other basins (for example please see: Pietkiewicz, M.; Klimach, A.; Ogryzek, M. Legal status of surface waters—comparative study on the example of lakes. Water 2020, 12, 326. (this source can be also useful to the argument above – legal definitions).

Authors writes In that in 1992, the State Council approved the establishment of the Tianjin Ancient Coast and Wetlands  National Nature Reserve  (verses 93-94) – the reference to the act should be given.

The article is of high value and after very small amendments in the introduction part should be published.

Author Response

Response to Reviewer 3 Comments

Dear Reviewer,
Thank your comments concerning our manuscript entitled “Dynamic Evaluation of Ecological Service Function Value of Qilihai Wetland in Tianjin” (ID: 933183). Those comments are all valuable and very helpful for revising and improving our paper, as well as the important guiding significance to our researches. We have studied comments carefully and have made correction which we hope meet with approval. Revised portion are marked in red in the paper. The main corrections in the paper and the responds to the reviewer’s comments are as flowing:

Responds to the your comments:

  1. Response to comment: For example, authors begin their elaboration with the definition of wetlands – this definition is of course theoretical, in reviewer opinion authors should give some legal definition of the meaning (please see for example Shine, C., & De Klemm, C. (1999). Wetlands, water, and the law: using law to advance wetland conservation and wise use (No. 38). IUCN. Afterwards, authors should give legal definition of wetland (if there is any) that is appropriate for the Tianjin region (or maybe for PRC).

Response 1:

Thank you for your advice. We added a legal definition of wetlands to the article.Including international and local legal definitions.We cite your suggested references and point out the definition of wetlands in China's wetland Protection and Management regulations

Modification: The definition of wetland in the "Wetland Convention" is considered to be the more authoritative concept of wetland "Wetlans are areas of marsh, fen, peatland or water, whether natural or artificial, permanent or temporary, with water that is static or flowing, fresh, brackish or salt, including areas of marine water the depth of which at low tide does not exceed six metres" (Shine, C., & De Klemm, C. (1999). Wetlands, water and the law: using law to advance wetland Conservation and wise use: IUCN.) In order to strengthen the management of wetland protection and fulfill the International Wetland Convention, China formulated regulations on wetland protection and management in 2013 and revised them in 2017. The revised term "wetland" refers to "perennial or seasonal water areas, water areas, and sea areas with a water depth of not more than 6 meters at low tide, including natural wetlands such as swamp wetlands, lake wetlands, river wetlands, coastal wetlands, and key protected wildlife Artificial wetlands such as habitats or key protected wild plant native sites"

  1. Response to comment: Authors writes about Sustainable Development Goals and quote appropriate literature – also in MDPI journals there are  a lot of possible refences to SDG’s when it comes to waters/lakes and other basins (for example please see: Pietkiewicz, M.; Klimach, A.; Ogryzek, M. Legal status of surface waters—comparative study on the example of lakes. Water 2020, 12, 326. (this source can be also useful to the argument above – legal definitions).

Response 2:

Thank you for your advice. We have added references to water/lake related articles on sustainable development.And the content of the article has been modified

"Based on the current achievement goals, the United Nations continues to guide the global sustainable development of a more comprehensive and specific 2030 Sustainable Development Goals (Spain). The Sustainable Development Goals 6 and 14 proposed by the United Nations are related to water, including "protection And restoration of water-related ecosystems.” A huge challenge currently facing the management of water-related ecosystems is that it is related to many human behaviors and environmental factors. Pietkiewicz, M.; Klimach, A.; Ogryzek, M. Legal status of surface waters—comparative study on the example of lakes. Water 2020, 12, 326. ) Therefore, it is of great significance to study the impact of human behavior on ecosystem changes. If relevant laws and management regulations can be formulated to maintain ecosystem functions or rebuild degraded ecosystems based on local actual conditions, then achieving the relevant sustainable development goals will much easier."

  1. Response to comment: Authors writes In that in 1992, the State Council approved the establishment of the Tianjin Ancient Coast and Wetlands  National Nature Reserve  (verses 93-94) – the reference to the act should be given.

Response 3:

Thank you for your comments! The article has revised this part of the content and added the bills involved in the establishment of nature reserves.

“On October 27, 1992, the State Council approved the "Request for Approval of the New National Nature Reserve in 1992" submitted by the Environmental Protection Agency (Guo Han [1992] No. 166), and approved the establishment of Tianjin Ancient Coast and Wetland National Nature Reserve.“

We tried our best to improve the manuscript and made some changes in the manuscript.  These changes will not influence the content and framework of the paper. And here we did not list the changes but they will marked in red in revised paper.

We appreciate for your warm work earnestly, and hope that the correction will meet with approval.Once again, thank you very much for your comments and suggestions.
